# Hyaluronan: Sources, Structure, Features and Applications

**DOI:** 10.3390/molecules29030739

**Published:** 2024-02-05

**Authors:** Katarína Valachová, Mohamed E. Hassan, Ladislav Šoltés

**Affiliations:** 1Centre of Experimental Medicine, Institute of Experimental Pharmacology and Toxicology, Slovak Academy of Sciences, Dúbravská cesta 9, 84104 Bratislava, Slovakia; 2Centre of Excellence, Encapsulation & Nanobiotechnology Group, Chemistry of Natural and Microbial Products Department, National Research Centre, El Behouth Street, Cairo 12622, Egypt

**Keywords:** chemical modifications, drug delivery, glycosaminoglycans, medical applications

## Abstract

Hyaluronan (HA) is a non-sulfated glycosaminoglycan that is present in a variety of body tissues and organs. Hyaluronan has a wide range of biological activities that are frequently influenced by molar mass; however, they also depend greatly on the source, purity, and kind of impurities in hyaluronan. High-molar-mass HA has anti-inflammatory, immunosuppressive, and antiangiogenic properties, while low-molar-mass HA has opposite properties. A number of chemical modifications have been performed to enhance the stability of HA and its applications in medical practice. Hyaluronan is widely applied in medicine, such as viscosupplementation, ophthalmology, otolaryngology, wound healing, cosmetics, and drug delivery. In this review, we summarized several medical applications of polymers based on the hyaluronan backbone.

## 1. Introduction

Any substance that is produced by cells or living organisms is called a biological molecule or biomolecule. Biomolecules have many structures and sizes and perform a wide range of functions. They are divided into four major types: lipids, proteins, nucleic acids, and carbohydrates [1].

Carbohydrates are among the most prevalent macromolecules on Earth and are structural elements of life and energy sources. Carbohydrates are mainly composed of molecules with atoms of carbon, hydrogen, and oxygen. The four types of sugar units that make them up are monosaccharides, disaccharides, oligosaccharides, and polysaccharides [2]. Polysaccharide, or glycan, can possess either a straight chain (a linear polysaccharide) or a branched molecular structure (a branched polysaccharide). Unlike mono-, di-, or some oligosaccharides, polysaccharides are not sweet in taste and are hydrophobic in nature. They do not form crystals, they have a high molar mass, they form a white powder, and the oxygen-to-hydrogen ratio is 1:2 [3]. These intricate biomolecules are both a structural element of plant cells and an essential source of energy for animal cells. Depending on the kind of structural monosaccharide units, polysaccharides are divided into homopolysaccharides or heteropolysaccharides. Heteropolysaccharides are composed of different types of monosaccharides, such as heparin, chondroitin-4-sulfate, and hyaluronan (HA) [4].

This review is focused on hyaluronan, especially its features and several medical applications. Hyaluronan is involved in nearly every aspect of biology. Cell and organ development, the body’s reaction to inflammation and tissue damage, cell migration, the development of cancer, and resistance to cancer are all impacted by its interactions with cell receptors or other extracellular binding partners. Hyaluronan has differing and sometimes contradictory effects on many biological functions, e.g., pro- or anti-inflammatory, promoting and inhibiting either migration or cell proliferation [5]. The greatest way to comprehend the biology of HA is to ascertain how each of its individual effects fits into a dynamic mechanism that governs biology at the cellular, organ, and organismal levels [6].

## 2. Hyaluronan

### 2.1. Structure and Properties of Hyaluronan

Meyer and Palmer were the first who, in 1934, isolated a material from the bovine vitreous body and purified it, and it was subsequently recognized as hyaluronan [7]. The HA chemical structure was discovered twenty years later. Hyaluronan is one of the three main β-chain sugars on Earth besides chitin and cellulose [8]. 

Hyaluronan is a monomeric mucopolysaccharide that is linear, negatively charged, water-soluble, non-sulfated, biodegradable, and viscoelastic. It is composed of two repeating units of saccharides, such as *N*-acetylglucosamine and d-glucuronic acid [9,10], which are linked alternately by β-1,4- and β-1,3-glycosidic bonds (Figure 1). The great flexibility and solubility of HA polymers are just due to the interspersed β-1,3 links. The average adult human body has 15 g of HA, of which 33% are replaced every day. In addition to having the simplest structure and not being covalently linked to a core protein, HA is the only glycosaminoglycan (GAG) not synthesized in the Golgi apparatus [11]. Hyaluronan is the major component of the extracellular matrix (ECM) of soft connective tissues, e.g., the skin, heart valves, umbilical cord, and vitreous body. Moreover, the brain, lymphatic, and synovial fluid (SF) contain large quantities of HA [12,13]. In ECM, HA serves as a scaffold for the binding of additional large GAGs and proteoglycans, which are preserved by particular HA–protein interactions [14]. Although HA also exists inside cells, its purpose is still unknown [11].

The length of the linear uncoiled HA polymer can be in the range of 10 nm–25 µm, and the number of disaccharide units is in the range of 25–25,000 [15]. The molar mass of HA is in the range of 100 kDa in serum to 8 MDa in the vitreous body [16]. One of the most amazing characteristics of HA is its ability to envelop itself in a substantial amount of water, up to several thousand times its initial weight [17]. When HA is in a solution, its rheological characteristics are dependent on molar mass, pH, ionic strength, concentration, and shear rate. High-molar-mass HA aqueous solutions are cohesive, viscous, and lubricating. It is well known that the viscosities of solutions containing high-molar-mass polymers at moderate concentrations are variable and decrease as the shear rate increases [18]. Aqueous solutions of HA display primary, secondary, and tertiary structures due to a variety of interactions involving hydrogen bonds within and between molecules.

In addition to being involved in various physiological processes such as wound healing, embryological development, repair, and regeneration, HA is also involved in proliferation, cell migration, differentiation, and adhesion in rapid tissue growth and numerous signal transduction pathways [19]. Hyaluronan chains of a medium size are linked to ovulation, embryogenesis, and wound healing; oligosaccharides with 15–50 repeating disaccharide units are immunostimulatory, inflammatory, and angiogenic; and smaller HA oligomers are antiapoptotic and inducers of heat shock proteins. In contrast, high-molar-mass HA polymers have antiangiogenic, immunosuppressive, anti-inflammatory, and space-filling properties [20]. Water homeostasis, osmotic pressure preservation, physiological solution buffering, joint lubrication, chondroprotection, and a variety of other bodily functions are regulated by HA. Hyaluronan also plays a role in space and volume expansion [21,22], immunological modulation, malignant transformation, and inflammation [23,24]. Moreover, HA levels are frequently correlated with the aggressiveness of human malignancies [25].

Hyaluronan is linked to significant phenomena that are essential to the quality of human life, such as increased longevity, cancer resistance, and newborn protection, which explains the long-term growth in interest in HA. Over the last 20 years (years 2002–2022), there has been an increase in the number of papers on hyaluronan/hyaluronic acid, which is summarized in Figure 2a. The types of publications on HA research within the last 20 years are denoted in Figure 2b, where almost 50% relate to research papers. 

### 2.2. Sources of Hyaluronan

Hyaluronan has been isolated from a variety of sources, including rooster combs, umbilical cords, and microorganisms such as Streptococci (Table 1) [26]. 

### 2.3. Modifications of Hyaluronan

Hyaluronan can be modified to produce a biomaterial that is more mechanically and chemically resilient while still being biocompatible and biodegradable. However, such a biopolymer also has drawbacks, such as poor mechanical properties and a short in vivo residence time caused by hyaluronidase-catalyzed degradation [22,30,31,32,33,34]. In order to eliminate these problems, different chemical alterations of HA were used to create new HA derivatives (HADs) with improved mechanical qualities and decreased susceptibility to chemical and enzymatic hydrolysis [35]. Cross-linking and esterification are two modification techniques used to change the mechanical and chemical properties of HA, which has allowed materials to be specifically tailored for use in drug delivery, cell adhesion, joint lubrication, and signaling. The most widely used HA derivatization method is supposed to be mediated by carbodiimides and utilized for hydrogel preparation and drug molecule attachment to couple water-soluble hydrazides to HA carboxylic acid groups at pH 4.75. It has also been documented that hydrazides can be used to functionalize alkanes and enhance the rheological properties of HA. Furthermore, researchers have examined the use of alkanes, silylation, and acylation to produce HA that is more hydrophobic to be molded hot or dissolved in organic solvents [36].

It is important to consider that when modifying HA, changes made to -COOH group can affect the behavior of the HA molecule in the body since HA receptors and hyaluronidase recognize carboxylic groups [37]. Hyaluronan functional groups (hydroxyl, carboxyl, and acetyl) can be modified chemically in two different ways: either by conjugation or cross-linking [38]. Conjugation involves a variety of chemical modifications, such as the introduction of special functional units including ether, ester, or amide groups; the attachment of bioactive as well as prodrug moieties; and the incorporation of marker molecules, such as specific dyes. The approach to chemically modify HA includes both conjugation and cross-linking processes [39], while cross-linked HAs, also known as hylans, are recognized for their extended residence time (up to 9 days) compared to native HA, which is degraded by specific HA enzymes, hyaluronidases (HYALs), both in situ and systemically [40,41], and extremely high molar mass (up to 23 × 10^6^ Da). Etherification (diglycidyl ether), bis-epoxide cross-linking, esterification, divynilsulfone cross-linking with epoxides, or dimethyl sulfone are common methods for changing HA hydroxyls. On the other hand, HA carboxyl can be changed by reactions mediated by carbodiimide, amidation, esterification, and dihydrazide, dialdehyde, or disulfide cross-linkers [42]. The preparation of new HA derivatives should take into account the multifunctionality of the HA molecule by selecting a mode of synthesis and applying mild reaction conditions. Special attention should also be paid to the properties of native HA, such as its excellent biocompatibility, adjustable biodegradability, and mucoadhesivity [39]. In order to achieve a delayed release and/or a prolonged effect, the majority of developments have concentrated on producing materials for implantation, such as films or sponges, and on combining HA with therapeutic agents to achieve a controlled release [43,44,45].

Cross-linking of HA can be performed by covalently joining tyramines (Corgel), methacrylates, and thiols (Extracel, HyStem). Formaldehyde, divinylsulfone, or formaldehyde can be used to cross-link HA to yield Hylan-A or Hylan-B [46]. Cross-linking with water-soluble carbodiimide, divinyl sulfone, polyvalent hydrazide, glutaraldehyde, disulfide, auto- and photo-cross-linking, and cross-linking with these hydrogels are in the process of being developed currently [47,48]. The cross-linking of HA increases its stability and prevents biodegradation of HA in vivo. Chemical modifications can be accomplished through the amidation of HA using homobifunctional reagents, which are new modifying reagents that have been synthesized and contain a new divalent disulfide-based protecting group. Amidation of HA with these reagents gives rise to either an one-end coupling product or intra- or intermolecular cross-linking of the HA chains [49], Ugi condensation (requiring four functional groups, such as primary amino, carboxyl, isocyanide, and aldehyde, for a typical Ugi condensation reaction) [50], ester formation (containing an irreversible ester bond that can be hydrolyzed by enzymes) [51], and ether formation (forming a thioethyl ether derivative of HA) [20].

## 3. Hyaluronan in Medical Practice

Hyaluronan solutions have amazing qualities such as hygroscopicity, non-thrombogenicity, non-immunogenicity, biodegradability, viscoelasticity, biocompatibility, and bioreactivity. These features enable their application in a variety of fields, such as the food, cosmetic, and biomedical industries [52].

### 3.1. Hyaluronan in Dermal Filler

In humans, about 50% of HA is found in the skin, i.e., 7–8 g per average adult human, with ~0.5 mg/g of HA found in the dermis and ~0.1 mg/g in the epidermis [53]. Recently, there has been a lot of interest in the field of cosmetic surgery for dermal rejuvenation due to the use of HA-based dermal fillers (DFs). Soft tissue augmentation with DF is now a standard procedure in aesthetics. Dermal fillers are used for facial rejuvenation and to temporarily eliminate the appearance of rhytides and skin folds [54]. Hyaluronan transports vital nutrients to these fibers and forms bonds with collagen and elastin. Due to the ability of HA to effectively augment soft tissue in a comprehensive nonsurgical facial rejuvenation approach, physicians utilize it to restore the durability of skin. A dermal filler exhibits clear antiaging benefits, such as a more youthful appearance, increased skin volume, elevated viscoelasticity, smoothness, improved lifting ability, and allow tighter skin [55].

Hyaluronan used to create DF is typically supplied as a dry powder. If this product is used as DF, drainage or degradation pathways would quickly remove it from the injection site. In order to overcome the inability of HA solutions to reside, a cross-linked HA can be used to create DFs. A weak gel is produced by cross-linking, which forms a network of polymers from a viscous solution. A chemical and physical barrier is added by the HA gel to prevent drainage or degradation. As a result, cross-linking HA lengthens residence time [56]. Since they have been in use for over 15 years, cross-linked HA fillers are thought to be well tolerated. They have good tissue integration, outstanding biocompatibility, and structural characteristics akin to native tissue. Cross-linking of HAD is necessary to prolong the duration of HA filler and stop it from being biodegraded by enzymes and free radicals. HA-based DFs differ in their physical and chemical characteristics, such as stress applied, particle size, cross-linking agent density, polymer concentration, extrusion force, and viscosity. HA-based DFs are made of bacterially fermented or rooster comb-derived HAs that have been cross-linked with butanedioldiglycidyl ether [57]. Dermal fillers must be elastic in a low-shear environment in order to be injected into the dermal connective tissue. It is suggested that greater persistence is caused by DF elasticity [58].

### 3.2. Hyaluronan in Tissue Engineering

The field of tissue engineering (TE) is broad and primarily concerned with the regeneration of lost or damaged tissues. By developing biological substitutes, TE allows for the replacement of damaged tissues both structurally and functionally while promoting tissue growth in vivo. These biological replacements are structures made by combining suitable cells and/or growth factors with scaffolds. The scaffold serves as a framework for developing tissues and cells and should function as a synthetic ECM to facilitate the attachment, migration, proliferation, and differentiation of cells until the creation of new tissue [59]. For TE applications such as soft tissue augmentation, wound healing, facial intradermal implants, and artificial skin, HA has demonstrated exceptional promise. Due to the high concentration of HA in the ECM, especially in nervous system tissue, HA has been utilized as the structural foundation for hydrogel scaffolds. Owing to its broad availability, nonimmunogenic properties, and ease of adjusting chain length, HA is a flexible macromolecule with enormous potential for use in TE applications. Hyaluronan hydrogels have the ability to exchange nutrients, oxygen, and metabolic waste while simulating the water content of individual tissues [60]. A variety of tissues, including skin, fibrocartilage, cartilage, intervertebral discs, bone, vascular tissues, adipose tissues, and heart valves, have been successfully regenerated using HADs and their blends as scaffolds. It has been established that the amides of HA are bioactive and appropriate carriers of cells for TE procedures [61].

### 3.3. Hyaluronan in Viscosupplementation

Degenerative joint disease, commonly referred to as osteoarthritis (OA), is the most prevalent type of arthritis. The financial toll that OA takes is substantial and multifaceted, encompassing expenses for medical care, hospital stays, missed work, and home care. With up to 41% involvement, the knee is the most commonly affected joint in OA, followed by the hands (30%) and hips (19%) [62]. In particular, the prevalence of knee OA has doubled since the middle of the 20th century. It has also been demonstrated that OA significantly lowers the quality of life. Viscosupplementation is the process of injecting HA into arthritic joints with the goal of restoring SF natural viscoelasticity and maintaining cartilage homeostasis and lubrication without inflammation [63,64]. One of the main functions of SF, a substance present in synovial joint cavities, is to reduce friction between the articular cartilages while the joint is in motion [65]. It has been observed that injecting HA preparations into patients with OA reduces their use of potentially hazardous nonsteroidal anti-inflammatory drugs and improves their symptoms [64].

Hyaluronan, coupled to a thermosensitive polymer, allowed for the spontaneous formation of nanoparticles at body temperature, thereby overcoming the drawbacks of repeatedly administered injections and the rapid breakdown of exogenous HA treatments. In addition to enabling the restoration of its mechanical characteristics, viscosupplementation also permits synoviocytes to be stimulated to produce high-molar-mass HA [66].

### 3.4. Hyaluronan in Ocular Treatment

A complicated and crippling inflammatory condition of the ocular surface is called dry eye disease (DED). Global epidemiological research indicates that the prevalence of DED varies between 5 and 50%. If it is extrapolated, there would be 400 million to 3.7 billion DED patients worldwide [67]. The prevalence of dry eye disease rises sharply after the age of fifty, with the condition becoming more common as people get older. DED creates significant financial burdens for both the individual patient and society at large by restricting participation in daily activities and the workforce and raising healthcare costs. Patients with DED experience a significant decline in their visual and overall quality of life. The healthy tear film coats and lubricates the ocular surface while serving as a chemical, physical, and immunological barrier against the outside world. An outer lipid layer and an inner mucoaqueous layer make up the tear film in a healthy eye, and they work together to maintain the stability of the ocular surface [68]. The initial line of treatment for DED is artificial tears. They shield the surface of the eyes and help stabilize and restore the tear film. This helps prevent further damage and reduce signs and symptoms by slowing or stopping the progression of DED.

On the market, there are varieties of artificial tears with different active ingredients. One often-used and clinically proven ingredient is hyaluronic acid [69]. Due to its abundance of hydroxyl groups, hyaluronic acid thickens and stabilizes the tear film, lubricates the ocular surface to reduce mechanical trauma, and aids in re-epithelialization. Additionally, hyaluronic acid lessens evaporation from the surface of the eyes, which is the primary cause of hyperosmolarity, one of the primary causes of inflammation and surface damage in DED [70]. In order to prolong the therapeutic effect of the eye-drop formulation, viscous HA solutions or hydrogels of higher concentrations are often used to slow down the drainage of the formulation. The primary benefit of HA solutions of high viscosity over polyvinyl alcohol or celluloses is their shear-thinning behavior. Hyaluronan solutions function as a general viscous mucoadhesive excipient, a lubricating and wetting agent, and a systemic drainage inhibitor for drug-containing eye drops, extending their ocular residence [71].

### 3.5. Hyaluronan in Wound Healing

Wound healing is an intrinsic field of study. In general, there are four main phases to normal wound healing: homeostasis, the inflammatory phase, the proliferative phase, and the remodeling phase. These phases share overlapping processes of coordinated cellular activity. These biological processes are mediated by cytokines, which enable the production of structural proteins and polymers by wound-healing cells [72,73]. Hyaluronan is involved in every phase of wound healing. There are different lengths of HA polymers, and each size of HA molecules has a specific function at a different phase of wound healing. In summary, large HA molecules regulate structure and occupy space; small HA fragments promote cell migration; medium-sized HA chains stimulate the expression of inflammatory cytokines; and very short, low-weight (four saccharides) HA fragments induce chemotaxis [74].

The inflammatory phase of wound healing is marked by a sharp rise in HA synthesis. Large, molecular-sized HA fragments are synthesized from platelets and are available in the bloodstream upon skin breach. By binding to fibrinogen, these HA fragments can start the extrinsic clotting process. Oedema develops at the wound site as a result of the significant volumes of HA released during wounding [75]. Due to the hydrophilic nature of HA, the tissue around the wound swells, allowing cells to move into the injured area and form a porous framework (Figure 3) [76].

### 3.6. Hyaluronan in Drug Delivery

The term “drug delivery” refers to technologies that transport medications into or through tissue. These technologies include delivery mechanisms such as an injectable vaccine or a pill that you swallow [77]. Drug delivery systems can also refer to the way that pharmaceuticals are enveloped, such as in a micelle or nanoparticle, to prevent degradation and enable the drug to go to the appropriate location within the body. The development of drug delivery has come a long way in the last few decades, and a more rapid advance is expected in the following years [78]. Biomedical engineering has made significant contributions to our knowledge of the physiological obstacles to effective drug delivery as well as to the creation of various novel drug delivery strategies that are now being used in clinical practice.

It has been discovered that the HA receptor CD44 is overexpressed in many tumor cells; however, it is expressed at a low level on the surface of epithelial, hematopoietic, and neuronal cells [79,80,81]. Consequently, the use of HA derivatives as drug carriers enhances drug targeting, transdermal absorption, sustained release, and thickening of the drug (Figure 4). When cytotoxic drugs are coupled with macromolecular substances, the drug’s pharmacokinetic profile is enhanced, its distribution is prolonged, and its elimination time is increased. Moreover, due to the drug’s delayed release from the carrier, the concentration of the medication in the plasma is reduced; however, it remains in the tumor tissue for a longer period of time. It has been shown that HA and drug conjugates have the dual benefits of receptor-mediated endocytosis and aggregation at the tumor site. Hyaluronic acid and its derivatives are applied in a variety of drug delivery systems, including cationic polymer gene carrier systems, drug delivery of gels [82,83], polyelectrolyte microcapsules, films [84], nanoparticles, etc. [85,86,87]. Since HA binds specifically to the receptors on the surface of cancer cells and is biodegradable and biocompatible, its use in targeted drug delivery of anticancer drugs has advanced significantly. It creates conjugates by reacting with other medications. The conjugates provide a targeted effect and controlled release to achieve the timing and directed release objectives. This makes it possible for them to deliver various medications to various pathological locations [88].

### 3.7. Hyaluronan in Transdermal Drug Delivery

Novel microneedle (MN) arrays made of HA were developed and used for transdermal insulin delivery [90]. It was proven that MNs could pierce the skin for at least an hour at 75% relative humidity. The fact that more than 90% of insulin in MNs remained in those reservoirs at all temperatures after they were stored for a month at a temperature in the range of −40–40 °C shows the high stability of insulin in MNs [91]. After being applied topically, these unique HA MNs have the ability to dissolve themselves, and insulin is released from them fast. Pharmacodynamic and pharmacokinetic parameters showed that insulin delivered via MNs was almost completely absorbed from the skin into the systemic circulation and that the hypoglycemic effect of insulin-loaded MNs was almost the same as that of insulin injected subcutaneously. Therefore, HA MNs could provide a secure and efficient transdermal insulin delivery system in clinical practice [92,93].

Bonet et al. (2023) [94] studied the therapeutic effect of HA when combined with each of three transdermal drug delivery enhancers (protamine, dimethyl sulfoxide (DMSO), or terpene) in preclinical models of inflammatory and neuropathic pain. The administration of a combination of HA with the transdermal enhancers (protamine and terpene) attenuated chemotherapy-induced painful peripheral neuropathy hyperalgesia rather than with the DMSO vehicle [95,96]. Also, Yuan et al. (2022) [97] prepared a hydrogel containing HA-modified transforms as a drug carrier for indomethacin (IND). Indomethacin was encapsulated in HA-modified transfersomes to enhance transdermal IND delivery and reduce adverse effects.

Son et al. (2017) [98] synthesized HA–dodecylamine and nanohydrogels with indocyanine green as a model agent and proved that nanohydrogels have a promising potential for applications as transdermal delivery systems in the pharmaceutical and cosmetic industries.

### 3.8. Hyaluronan in Nasal Drug Delivery

The intranasal administration of molecules has been investigated as a non-invasive way for the delivery of drugs to the brain in the last decade. Circumvention of both the blood–brain barrier and first-pass elimination by the liver and gastrointestinal tract are considered the main advantages of this method. Due to the rapid mucociliary clearance in the nasal cavity, bioadhesive formulations are needed for effective targeting. Hyaluronan is used as a mucoadhesive in a nasal formulation, increasing the brain penetration of a hydrophilic compound in the size of a peptide via the nasal route [99,100,101]. The blood-cerebrospinal fluid (BCSF) and blood–brain barriers (BBB) make it difficult to target medications to the central nervous system (CNS). Many techniques have been tried to improve blood–brain transport and deliver medications to the brain at effective concentrations [102]. Prodrugs, drug binding to transferrin or targeted vesicle systems, hyperosmotic shock, vasoactive substances, and inhibition of efflux transporters are some of the strategies used to manipulate the BBB (Figure 5). There are additional methods to improve drug transport to the central nervous system, though, aside from changing the drug molecule or the BBB’s functioning. Selecting an application site that stays aside from the BBB is one such strategy. The delivery of medicine to the brain through the olfactory region through the nasal route has garnered a lot of attention lately [101,102,103]. In a distal epithelium, olfactory sensory neurons are the only first-order neurons with cell bodies. For the distinct anatomical and physiological characteristics of the olfactory region, the direct exposure of dendritic processes to the external environment in the upper nasal passage permits both extracellular and intracellular pathways into the CNS that circumvent the BBB [104].

Intranasal administration is noninvasive, virtually painless, and finally simple for patients as well as physicians to perform in emergency situations. It also provides an immediate starting point for therapeutic effects and eliminates the first-pass effect or gastrointestinal drug degradation [105]. Moreover, it guarantees fast absorption, prevents first-pass metabolism in the liver and gut, and does not need to be sterile. Hyaluronan was shown to prevent chronic rhinosinusitis [106].

Horvát et al. (2009) [99] developed a formulation containing sodium hyaluronate in combination with a non-ionic surfactant to enhance the delivery of hydrophilic compounds to the brain via the olfactory route. They used fluorescence isothiocyanate-labeled 4 kDa dextran (FD-4) as a test molecule. Hyaluronan increased the viscosity of the vehicles and slowed down the in vitro release of FD-4. On the other hand, Laffleur et al. (2023) [107] modified HA with cysteine ethyl ester hydrochloride via amid bond formation, and a strongly increased bonding to the mucosa by the formation of covalent disulfide bridges could be observed in comparison to unmodified HA. The adhesion study with the model drug showed a more pronounced binding to the nasal mucosa in the case of modified HA in comparison to the unmodified one. Casula et al. (2021) [108] recently examined a natural and green nasal spray containing hyaluronic acid and the extract of *Zingiber officinalis* as a novel strategy for the treatment of rhinitis and rhinosinusitis.

### 3.9. Hyaluronan in Colon Drug Delivery

Colorectal cancer (CRC) is the fourth most common cause of cancer death around the world. Due to increased life expectancy and a diet rich in fat and low in fiber, the prevalence has been rapidly increasing. The primary method of treatment for CRC management is chemotherapy. The clinic uses a number of FDA-approved the olfactory chemotherapeutic medications, such as capecitabine, irinotecan hydrochloride, and fluorouracil [109]. There is a pressing need for advanced colon-specific local drug delivery systems that can provide major advantages in treating diseases associated with the colon, such as inflammatory bowel disease and colon cancer. A precise, colon-targeted drug delivery platform is expected to reduce drug side effects and increase the therapeutic response at the intended disease site locally [110]. Kotla et al. (2019) [111] reported a strategy that uses a HA-functionalized polymeric hybrid nanoconjugate system fabricated from processes that are generally recognized as safe reagents for enhanced drug absorption with controlled drug release and increasing local drug bioavailability in the colon lumen. They functionalized the HA polymeric hybrid nanoparticulate system, curcumin-HA NPs, whereas curcumin was used as a model fluorescent drug.

Liposomes are a type of nanocarrier for drug delivery that have been studied in detail and have many applications. Liposomes have the ability to control drug release synchronously, reduce drug accumulation in tumor sites, and lessen drug toxicity to cells other than cancerous ones. Moreover, since liposomes have a hydrophilic core surrounded by a lipid bilayer, they serve well as delivery systems for both lipophilic and hydrophilic drugs [112]. Active targeting of liposomal nanocarriers to the intended site of action can significantly increase their efficacy. Active targeting strategies for anti-cancer drugs investigate specific interactions between liposome-bound targeting groups and cell surface receptors. Moreover, the majority of cancerous cell types overexpress hyaluronan receptors to raise the concentration of HA in the surrounding tissue of tumors. Additionally, colorectal cancer has overexpressed hyaluronan receptor isoforms.

As a result, cancerous cells with the greatest potential for metastasis usually exhibit enhanced HA binding and internalization. Hence, the HA receptors overexpressed on colon cancer cells are more readily bonded to the HA-conjugated liposomes encasing both medications, cell-specific targeting is possible [113].

### 3.10. Hyaluronan in Chemoprevention and Cancer Therapy

It has been reported that HA significantly alters a number of signaling pathways linked to cancer. The majority of its effects are mediated by interactions with cell surface receptors (TLR-2, TLR-4, RHAMM, LYVE, and CD44), which bind and react to HA when they are activated. The main ligand for CD44 is HA. CD44 is overexpressed in a variety of malignancies, such as pancreatic, breast, lung, ovarian, prostate, and so forth. Transmembrane glycoproteins belonging to the CD44 family are important regulators of signal transduction [114]. As HA affects oncogenic signaling pathways, scientists have discovered effective ways to block hyaluronan-receptor interactions to impede the growth of tumors. Treating with hyaluronan oligomers that compete with polymeric hyaluronan for constitutive binding to its endogenous receptors appears to be one strategy. This approach is predicated on the idea that high-affinity, multivalent interactions with receptors are necessary for endogenous HA to transduce signals; small oligomers that bind monovalently will act as antagonists by acting as a substitute for cooperative, multivalent interactions with low-affinity and low-valency receptors [115]. Combining a drug with HA not only provides pharmacological benefits such as stabilization and solubilization, but it also confers selectivity and specificity for cancerous cells. Together, these data suggest that the use of HA, either itself or in conjunction with well-known medications (taxol, doxorubicin, methotrexate, vincristine, 5-fluorouracil, gemcitabine, imatinib, cisplatin, etc.), could be a promising option for the treatment of cancer [116,117].

## 4. Conclusions

Hyaluronan, an outstanding material, has established a strong foundation for biomedical applications due to its remarkable multifunctional capabilities. In order to build new platforms for various applications, hyaluronan has developed a new requirement through derivatives or grafting modification. Researchers have become more interested in using it in various biomedical applications, including drug delivery and regenerative medicine, due to its exceptional biodegradability, biocompatibility, ease of chemical functionalization, and distinctive physical, biological, and chemical properties.

## Figures and Tables

**Figure 1 molecules-29-00739-f001:**
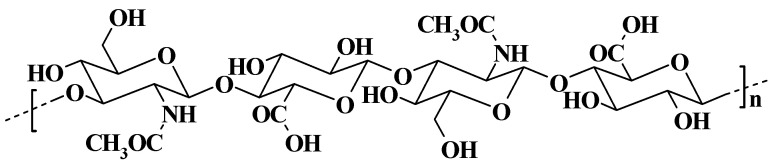
Hyaluronic acid chemical structure.

**Figure 2 molecules-29-00739-f002:**
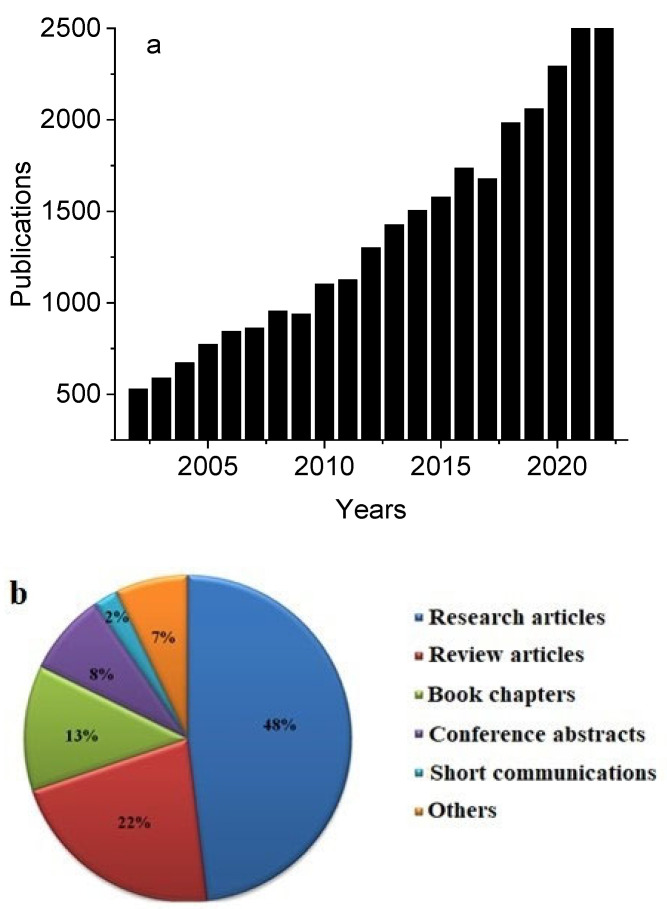
(**a**) Annual number of HA articles indexed in Pubmed database over the last 20 years. (**b**) Types of HA articles indexed in Science Direct database during the last 20 years.

**Figure 3 molecules-29-00739-f003:**
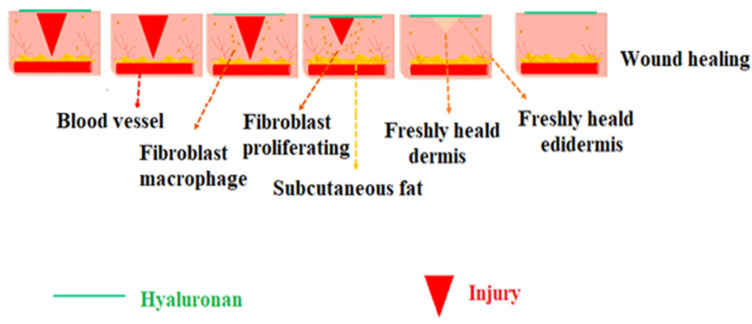
The impact of HA at wound sites throughout the stages of wound healing includes hemostasis, inflammation, proliferation, and remodeling (adapted from Sudhakar et al. [76]).

**Figure 4 molecules-29-00739-f004:**
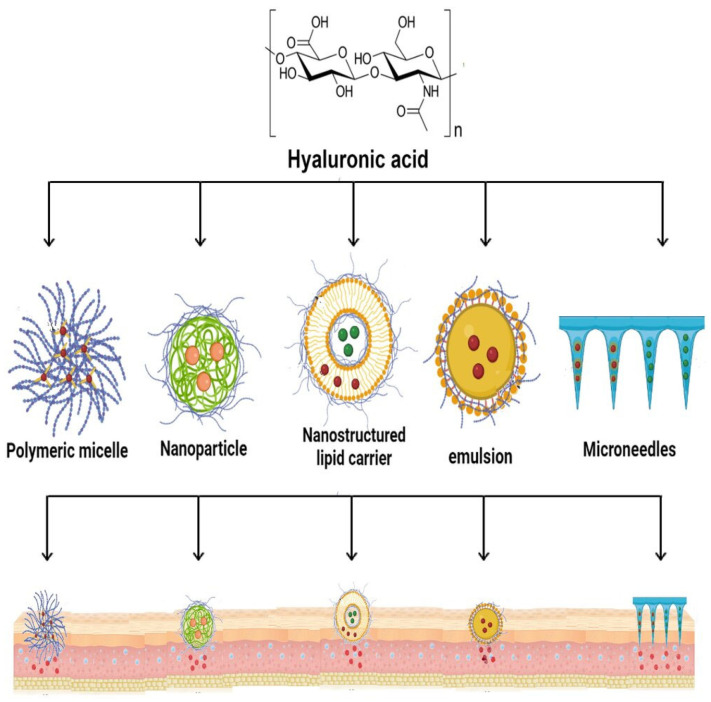
Drug delivery formulations using HA and its derivatives (adapted from Juhaščik et al. [89]).

**Figure 5 molecules-29-00739-f005:**
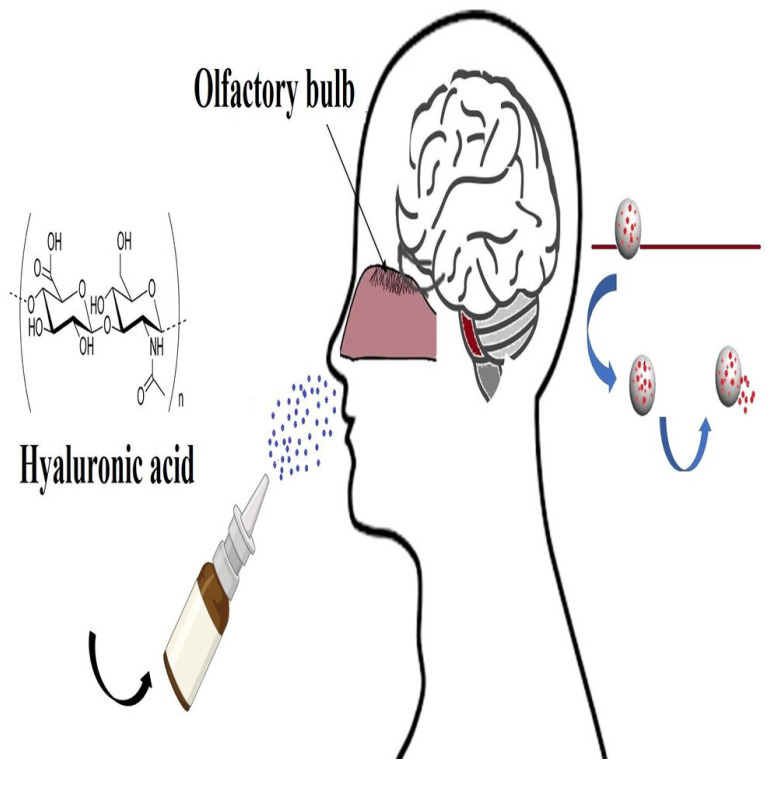
Application of hyaluronan in nasal drug delivery.

**Table 1 molecules-29-00739-t001:** Occurrence of hyaluronan in living organisms.

Source	Occurrence	Reference
Humans	Urine, epidermis, dermis, serum, vitreous body, umbilical cord, synovial fluid	[27,28]
Bacteria	Certain bacteria capsules (such as those of streptococci)	[26]
Roosters	Rooster comb	[19]
Sheep	Synovial fluid, medulla cortex, lungs	[13]
Rabbits	Renal papillae, kidney, vitreous body, muscle, liver	[14]
Rats	Lung, kidney, brain, liver	[14]
Cattle	Bovine nasal cartilage	[29]

## Data Availability

Not applicable.

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
