# Peer review of "Hyaluronan: Sources, Structure, Features and Applications"

_molecules, 2024, doi:10.3390/molecules29030739_

Round 1

Reviewer 1 Report

Comments and Suggestions for Authors

The paper is a review of the use of hyaluronic acid in biomedical applications. It is a well written, easy to read review on the subject. 

References are satisfactory and comprise other review papers to a large extent.

Some minor comments:

Figure 1: change place on subfigure description glucuronic acid and acetylglucoseamine. 

Line 122: "Increased residence time" should be specified. Increased as compared to what? What residence time is increased?

Line 130: "Replacing HA" should this not be "combine" HA with therapeutic agents to achieve controlled release. Otherwise this statement needs better explanation.

Line 280: "Hyaluronan is the typical carrier used in drug delivery....." I guess this is meant as compared to hyaluronan derivatives. Hyaluronan is not the typical carrier in drug delivery in general.

Paragraph 3.7, 3.8 and 3.9. Different areas of drug delivery are described in general. Use of HA is only exemplified with one example. It would be nice to read more about examples. I would also suggest to combine the drug delivery areas in a single chapter and reduce the description of the drug delivery area. 

Line 333: use of the olfactory route as a drug delivery alternative is not new. The indicated renewed attention should be confirmed with appropriate references.

Author Response

The answers to reviewer's comments are attached.

Reviewer 2 Report

Comments and Suggestions for Authors

Reviewer’s report on the manuscript entitled, “Hyaluronan: Sources, Structure, Features and Applications” by Katarína Valachová *, Mohamed Hassan, Ladislav Šoltés, (Manuscript ID: molecules-2785834)

 The title of this review suggests a broad focus of information on hyaluronic acid (hyaluronan, HA). Indeed, the review presents a wide range of summaries on general aspects of HA as the title states, but also includes modifications and a large section with several sub-sections on the specific medical aspects and applications of HA. However, when reading the abstract (line 12) a reader will likely presume that that there is a particular emphasis on chemical modification of hyaluronon (HA). However, the larger portion of the review summarizes medicinally related applications.

 Authors have given a brief overview of HA, its benefits, and a few representative applications. This review is rather brief, given the breadth of the literature on HA and the general interest in this polysaccharide. Nonetheless, the current review is basically useful – in part because brevity. Readers whose interest is piqued have sources provided with which to begin a deeper look into more literature.

 While the review presents a summary of research areas that have already been described in the literature, one point that is not addressed is to assess (potential) future directions – a sort of future trends and perspectives discussion. Although this reviewer wishes to not infringe on the authors’ goals for the review, it would be helpful to readers if more details were presented alongside of the broader, more general review on HA. To illustrate, a few examples are seen in:

Line 100 – what does “mechanically unstable” specifically mean or include? What constitutes “unstable”? I appreciate the references but of what value is this review if only to have to go track down what this term implies? A simple search outright is easy enough to get plenty of literature on this aspect.

Line 126-127 is a completely vague statement. There is nothing to learn from this – what labs? What special qualities? Even just a few specific examples would be helpful here to increase the value of this review article.

So where possible, additional specifics and descriptions could be included to increase the value of the review.

  It is a bit surprising that only 89 references are cited, some of which are ~20 years old. While that alone doesn’t negate their value, the current state of HA research is of greater interest and value. In most cases, the authors cite only a single reference for points presented. For consideration, it appears that perhaps there are many other articles that could be included to more broadly represent the work done with HA on the selected topics presented here – see for example the authors’ review in Int. J. Mol. Sci., 2021, 7077.

 Two specific points to consider are:

 1.       The introduction is weak, just over a single paragraph, and reads too much like an introductory textbook entry. Perhaps this is the authors’ intent, but it seems that the information is too basic. HA is only mentioned on the last line as an example of a monosaccharide. It is suggested that the introduction could be improved by including a focused paragraph on the importance of HA itself, which is then interspersed and further highlighted throughout the sub-sections of Section 3.

2.       In Figure 1, the structural names are reversed. N-acetylglucosamine is the left side structure. The linkage identifications also need to be changed if the structure is to be presented in this fashion. It is worth stating that at physiological pH, the acid is deprotonated to form the hyaluronate ion. The Figure 1 caption should define the conditions under which the molecular state is represented.

Author Response

(The authors gave the same response as above.)
